# A Fragmenting Protocol with Explicit Hydration for Calculation of Binding Enthalpies of Target-Ligand Complexes at a Quantum Mechanical Level

**DOI:** 10.3390/ijms20184384

**Published:** 2019-09-06

**Authors:** István Horváth, Norbert Jeszenői, Mónika Bálint, Gábor Paragi, Csaba Hetényi

**Affiliations:** 1Chemistry Doctoral School, University of Szeged, Dugonics tér 13, 6720 Szeged, Hungary; 2Institute of Physiology, Medical School, University of Pécs, Szigeti út 12, 7624 Pécs, Hungary; 3Department of Pharmacology and Pharmacotherapy, Medical School, University of Pécs, Szigeti út 12, 7624 Pécs, Hungary; 4MTA-SZTE Biomimetic Systems Research Group, Dóm tér 8, 6720 Szeged, Hungary; 5Institute of Physics, University of Pécs, Ifjúság útja 6, 7624 Pécs, Hungary

**Keywords:** peptide, interaction, design, affinity, optimization, binding, water, structure, correlation

## Abstract

Optimization of the enthalpy component of binding thermodynamics of drug candidates is a successful pathway of rational molecular design. However, the large size and missing hydration structure of target-ligand complexes often hinder such optimizations with quantum mechanical (QM) methods. At the same time, QM calculations are often necessitated for proper handling of electronic effects. To overcome the above problems, and help the QM design of new drugs, a protocol is introduced for atomic level determination of hydration structure and extraction of structures of target-ligand complex interfaces. The protocol is a combination of a previously published program MobyWat, an engine for assigning explicit water positions, and Fragmenter, a new tool for optimal fragmentation of protein targets. The protocol fostered a series of fast calculations of ligand binding enthalpies at the semi-empirical QM level. Ligands of diverse chemistry ranging from small aromatic compounds up to a large peptide helix of a molecular weight of 3000 targeting a leukemia protein were selected for systematic investigations. Comparison of various combinations of implicit and explicit water models demonstrated that the presence of accurately predicted explicit water molecules in the complex interface considerably improved the agreement with experimental results. A single scaling factor was derived for conversion of QM reaction heats into binding enthalpy values. The factor links molecular structure with binding thermodynamics via QM calculations. The new protocol and scaling factor will help automated optimization of binding enthalpy in future molecular design projects.

## 1. Introduction

Determination of structure and binding thermodynamics of target-ligand complexes is a key step in drug design [1]. Thermodynamic quantities can be measured by experimental methods such as isothermal titration calorimetry (ITC [2,3,4,5,6,7,8,9,10,11]). Experimental measurements are often restricted by the lack and high cost of pure and concentrated target (protein) samples. Molecular structures and binding thermodynamics can be also predicted [12,13,14,15,16] by fast and cheap molecular mechanics methods. At the same time, molecular mechanics has serious limitations of calculation of electronic effects in complex structures. Such effects are present in almost all intermolecular interactions including ‘exotic’ cases such as cation-π interactions between aromatic and charged side-chains [4,17] or polarization effects at structural water molecules [18]. Quantum mechanical (QM) approaches can properly handle electronic effects of intermolecular interactions. However, hydration and large size of target-ligand complexes impose further challenges on QM methods as detailed in the following paragraphs.

Hydration largely affects the structure and function of various biomolecules and their complexes [19,20]. Water molecules of the complex interface contribute to the stability and specificity of target-ligand interactions [21,22,23,24,25,26,27,28] by building hydrogen bonding networks [29,30], restraining interatomic distances, and filling cavities [19,31]. Despite their importance, determination of positions of interfacial water molecules is not trivial [32]. Available water positions have been determined mostly [33] by X-ray crystallography. However, even this well-established technique suffers from numerous limitations. Assignation of electron density peaks to possible interface water positions is still not a routine job due to inherent mobility of water and large number of degrees of freedom [34] and the quality of the structure depends on the solute size [35]. Protein hydration in the crystal is not the same as in solution [36] which is further complicated by cryo-artefacts [36]. Overfitting of electron density data and misleading identification of water sites were found to be a bad practice [25]. Other experimental techniques such as nuclear magnetic resonance spectroscopy or cryo-electron microscopy have produced a relatively small number of structures with water positions assigned. To overcome the above limitations of experimental methods, theoretical approaches were developed to help the assignation of water positions. These approaches either assign water positions based solely on solute structures [37] or involve calculation of dynamics [38,39,40,41,42,43,44,45] of water–water interactions. In the present study, a molecular dynamics-based method MobyWat [32,46] will be applied for completion of hydration structures of target-ligand interfaces.

Besides hydration, system size is another challenge of calculation of large complexes at the QM level. Such investigations would require large computer resources if the entire target molecule was calculated. A decomposition of the target-ligand complex into tractable sub-systems can handle this problem. There are at least two approaches to conduct such a decomposition. The first approach applies QM for the binding site and molecular mechanics simulations to the rest of the system [47,48,49,50,51,52]. Another branch of methods is based on skillful fragmentation of the target and applies QM for the sub-system of target fragments and the ligand. For example, Zhang and Zhang [53] developed a method for molecular fractionation where the protein is decomposed into individual capped fragments. They performed ab initio HF and DFT QM calculations for the target-ligand complexes. Nikitina et al. [54,55] cut the heavy atoms of the target at a distance equal or less than 5 Å from any heavy atom of the ligand. They also used structural water molecules determined by X-ray analysis, inserted new ones according to H-bonding valences of the solute molecules [54] and also proposed an iterative scheme [55] of in silico hydration. They developed correlations for binding enthalpy (ΔH_b_) on sets of 8 [54], and 12 [55] complexes, respectively. The complexes included protein targets with small ligands of molecular weight (MW) up to 700 and the calculations were conducted at semi-empirical QM level using the PM3 parametrization. Dobes, Hobza et al. [56] investigated the small-molecule purine inhibitor Roscovitine in complex with cyclin-dependent kinase 2 at B3LYP/6–31G** and MP2 levels of theory. They cut the chains of the kinase target into small fragments of a few amino acids at the C_α_-N bond. The peptide bond was maintained and they considered only amino acids and crystal water molecules located within 5 Å from the ligand.

Structure-based calculation of thermodynamic properties such as ΔH_b_ is a central issue of engineering of efficient drug candidates. Enthalpic optimization of new lead molecules [57,58,59] is a successful pathway of drug design and requires determination or prediction of ΔH_b_ of target-ligand complexes. Despite the need for ΔH_b_ data, there are only a few QM studies on fragment-based calculation of target-ligand binding thermodynamics. Available studies of the previous paragraph mostly work with ligand molecules of moderate size. Complexes of large (peptidic) ligands with numerous hydration sites have not been studied extensively. Moreover, development of automated tools for extraction of structures of complex interfaces and a reliable hydration scheme would be also helpful for such fragment-based QM investigations.

A new protocol was introduced and tested in the present study to help the enthalpic design of drug candidates by answering the above challenges of automation of structure-based calculation of complexes of large ligands. For this purpose, an end-point approach was adopted for the calculation of ΔH_b_ according to Equations (1) and (2). As the reaction occurs in a biological environment, T and L water molecules hydrate the target and the ligand, respectively. Waters can also remain bound to the partners (s = 0), join the complex from the surrounding bulk (s > 0) or leave (s < 0) during ligand binding. The reaction heat (Δ_r_H) of the binding process of Equation (1) can be calculated [14,15,54,55,60,61,62] according to Hess’s law (Equation (2)), where Δ_f_H represents the calculated heat of formation of a reactant or a product as indicated in brackets.
Target[H_2_O]_T_ + Ligand[H_2_O]_L_ + s H_2_O = Target:Ligand[H_2_O]_T+L+s_(1)
Δ_r_H = Δ_f_H(Target:Ligand[H_2_O]_s_) − Δ_f_H(Target) − Δ_f_H(Ligand) − sΔ_f_H(H_2_O)(2)

This end-point approach is simple and it has been successfully applied in previous publications [14,15,54,55,60,61,62]. In the present study, it was particularly useful for screening of various solvent models and conducting several trials in reasonable time. In the forthcoming sections, the fine-tuning of the corresponding protocol, and the development of a relationship between calculated reaction heats and experimental binding enthalpy values will be described.

## 2. Results and Discussion

### 2.1. Fragmenter

As it was discussed in the Introduction, involving the entire target structure in a QM calculation is not feasible within a reasonable calculation time. Thus, QM calculation of the above Δ_f_H values (Equation (2)) necessitates an extraction of the interface region of the target-ligand complex. However, extraction of the complex interface and automated fragmenting of the target protein has no trivial solution. In the present study, a new protocol was elaborated including a fragmentation method, Fragmenter, to standardize the extraction of target-ligand interfaces (Figure 1). Fragmenter works on a complex structure including a target, a ligand and several water molecules. Amino acids of fragments are selected according to their intermolecular distance cut-off (d_TL_, Table 1). A brief overview of Fragmenter and the data stream are sketched in Appendix A and technical details are provided in Methods.

Fragmenter focuses on the neighboring parts of the target protein which have considerable interactions with the ligand and the interfacial water molecules. The whole ligand molecule and protein residues of interface regions of the complexes are extracted. The residues of the target molecule are preferably extracted as peptide fragments instead of single amino acids. The main goal is to obtain the shortest but continuous peptide chains from the target protein in a standardized way.

Thus, there is still a benefit of a considerably reduced target part, and continuity is also kept wherever it is possible. Parameter n specifies how many adjacent amino acids are added to the fragment chain of amino acids extracted according to d_TL_. After some experimenting (Appendix A), it was found that n = 0 produces good correlations (as seen in the following sections), and it was not necessary to investigate n = 1 for the systems of the present study. Fragmenter was implemented as a free web service (Appendix A). It provides the extracted complex interface structure (target fragments, ligand and water molecules) as an interactive image, also downloadable as PDB and Mopac input files from the ‘results’ tab (Appendix A) and also displays a list of estimates of per-residue intermolecular interaction energy (E_inter_) values to indicate unwanted close contacts.

### 2.2. Dry Systems and an Implicit Water Model

Having the Fragmenter protocol developed and implemented, Δ_f_H calculations of the (hydrated) target-ligand complex interfaces were conducted in a simplified and standardized way. Fragmenter was applied on all systems of Table 1 for extraction of the complex interfaces. All systems were prepared for Fragmenter using standard molecular mechanics energy minimization and explicit hydration protocols as described in Methods. The Δ_f_H values were calculated for the individual reactant (ligand and target fragments) and product (complex interface) structures, respectively. The calculations were performed at semi-empirical level using PM7 parameterization, with and without the Mozyme approach (Methods). The resulted, raw energy values are listed in Appendix A.

Within the end-point approach (Introduction), calculation of Δ_f_H of the reaction participants (Equation (2)) and a linear scaling (Equation (3)) of Δ_r_H to known experimental ΔH_b_(exp) values is necessary for calculation of ΔH_b_.
ΔH_b_(exp)_i_ = αΔ_r_H_i_ + β + ε_i_ = ΔH_b_(calc)_i_ + ε_i_, where i = 1, 2, …, N(3)

In the present study, 15 systems (N = 15) of Table 1 were involved in the derivation of regression coefficients (α, β) yielding ΔH_b_(calc) values and residuals (ε). Statistical parameters obtained for the dry complexes and various solvent models are listed in Table 2. Nine of the 15 systems with small ligands up to a MW of 550 were considered in a previous paper [55] as well. In the present study, additional six systems with large peptide ligands were included in the set as they often impose a challenge during lead optimizations due to their size and extensive hydration. Thus, the set of 15 systems involves various ligands with MW up to 3318, two orders of magnitude larger than the previous set. The experimental ΔH_b_ values cover a wide range between −2.935 and −15.5 kcal/mol (Table 1).

In the first step of the present investigations, no solvent models were applied (s = 0 in Equations (1) and (2)). That is, dry input structures without explicit water molecules were calculated in vacuo. The complete lack of water models resulted no correlation between the calculated and experimental ΔH_b_ values (column Vacuum/Dry in Table 2, Figure 2). The application of an implicit water model (COnductor-like Screening MOdel, COSMO [63]) increased the correlation (column COSMO/Dry in Table 2). However, this correlation can still be improved as reflected by the cross-validation. In general, the use of COSMO proved advantageous if compared with the vacuum/dry results (Table 2). There was a single case of System 2ke1 where ΔH_b_(exp) could not be converted to 298.15 K and the original value at 296.15 K (Appendix A) was used for the regressions of Table 2. To check the influence of this data point on the results, linear regressions were performed without System 2ke1, as well. The statistical parameters showed (Appendix A) that leaving out System 2ke1 did not improve the results in vacuo and COSMO yields considerable correlation.

### 2.3. Explicit Hydration and a Hybrid Model

A systematic investigation on explicit hydration was conducted to further improve the correlations of the previous section. It is challenging to give a straightforward definition for the origin of water molecules in the complexes, and prediction of ligand-bound water molecules is rather uncertain due to the relatively small binding surface of ligands. Thus, T = L = 0 was set and all interface water molecules were considered as if they had originated from the surrounding bulk solvent (s > 0, in Equations (1) and (2)). Hydration structure of the target-ligand complex was built up by the MobyWat method [32] and extracted by Fragmenter as part of the interfaces. MobyWat can produce complete, void-free hydration structures of complex interfaces. This is guaranteed by a soaking step during the systematic evaluation of a series of snapshots of molecular dynamics simulations accounting for water–water interactions besides solute-water ones. Thus, MobyWat can find all experimental reference water positions in many cases [32] and assign water positions not detectable by experimental [25,33,34,64,65] measurements.

In the present study, three shells were defined according to d_w_ (Table 1 and Appendix A) using interfacial water molecules (Figure 3A). Shell 1 contains water molecules closest to the solutes (d_w_ = 3.5 Å). Shell 2 holds waters with intermediate positions (3.5 Å < d_w_ < 5.0 Å). Shell 3 consists of all interfacial water molecules of Shells 1 and 2 with a d_w_ = 5.0 Å.

The use of explicit water molecules in vacuum improved the ‘dry’ correlations to an R^2^ of 0.44 (all systems, column Vacuum/Shell 3 in Table 2) and 0.65 (without System 2ke1, Appendix A). Cross-validation indicates that this improvement of correlation is robust only without System 2ke1. At this point, it seemed reasonable to check whether a hybrid model using both implicit and explicit hydration further improves the correlation. Indeed, the hybrid model (column COSMO/Shell 3 in Table 2) provided the best agreement between calculated and experimental ΔH_b_ with an *R*^2^ of 0.73 (Figure 2) using all interfacial water molecules. Notably, Shell 1 waters also yielded considerable correlation (*R*^2^ of 0.65). In both cases, the correlations survived the challenge of cross-validation. The intermediate water positions alone (Shell 2) yielded a stable correlation only without System 2ke1 (Appendix A).

To investigate the effect of ligand size and target diversity on the stability of the above correlation (COSMO/Shell 3), the set of systems in Table 1 was split into two sub-sets according to ligand MW. The first sub-set contains nine systems with small ligands of MW < 600. All these ligands have a common target, beta trypsin. The second sub-set contained six systems with large ligands of MW > 1000 and various targets. Linear regressions were performed separately for the two sub-sets and ΔH_b_(calc) values were calculated by the two regression equations, respectively. Overall statistical parameters obtained (Appendix A) were comparable to those of the regression for all systems (column COSMO/Shell 3 in Table 2) detailed above. Thus, stability of the correlations is not influenced by ligand size and target diversity of the systems in the case of the hybrid model.

### 2.4. Scaling Factor

The above COSMO/Shell 3 model with β ≠ 0 in Equation (3) is significant and robust regarding its overall regression parameters. However, the t_β_ value (Table 2) indicates that the level of significance of regression coefficient β is moderate (*p* = 0.015). Thus, a linear regression with β = 0 was also developed and the corresponding statistical parameters are listed in the last column of Table 2 (Appendix A). In this way, a model of high significance (*p* < 0.01) of all parameters was obtained and Equation (3) was simplified. The resulting Equation (4) includes only the value of regression coefficient α, which serves as a single, unit-independent scaling factor for conversion of calculated Δ_r_H into ΔH_b_.
ΔH_b_ = 0.031 (±0.002) Δ_r_H(4)

A similar value of 0.032 (±0.002) was obtained for the scaling factor if System 2ke1 was not involved in the regression. Via QM calculations, this factor serves as a direct link between molecular structure and binding thermodynamics of molecular complexes.

### 2.5. Case Studies on Hydration Structures

In two-thirds of the 15 systems, application of Shell 1 or 3 explicit water molecules resulted in the decrease of residuals (COSMO models in Table 2). Shell 2 waters have similar effect in one-third of the cases. For example, in the case of System 1k1l, the residuals decreased from 2.59 (dry) to 0.43 (Shell 3, β ≠ 0) and 1.74 kcal/mol (Shell 3, β = 0, Table 2), and a similar trend can be observed for the vacuum values.

In the interface of System 1k1l extracted after molecular mechanics energy-minimization (Figure 3A), Shells 1 and 2 contain 5 and 10 water molecules, respectively (Table 1). The water molecules of Shell 1 (Figure 3A) are located at the bottom of the interface bridging between the target and ligand (solute) partners. Shell 2 waters mostly occur at the opening of the interface towards the bulk (right side of Figure 3A) waters/region. As it was expected, large clusters of waters gathered around charged or polar groups. For example, the sulfonyl group (Figure 3B) of the ligand is surrounded by a group of water molecules, and only one of them belongs to Shell 1. No interactions were observed between the waters and the closest target fragment (G_216_SG_218_).

During semi-empirical QM relaxation (Figure 3C), positions and orientations of some water molecules were changed. For example, two water molecules (marked with crosses in Figure 3C) were shifted by 3.2 and 1.8 Å. The orientation of Shell 1 water molecule (marked with asterisk in Figure 3C) was changed to interact with the target fragment. Such changes resulted in an extensive H-bonding network of water molecules stabilizing the target-ligand interaction around the sulfonyl group. Formation of new hydrogen bonds imply that some of the Shell 2 water molecules became Shell 1 (not marked in Figure 3C).

While the hydration structure underwent a remarkable transformation during semi-empirical QM relaxation, the conformation of the target fragment was preserved. The above example of System 1k1l (Figure 3) showed how water molecules in the different shells contribute to the completion of the target-ligand interface structure and a consequent decrease in residuals of calculated ΔH_b_.

Besides small, rigid ligands like the phenylalanine derivative of System 1k1l, large peptide ligands were also involved in the present study. For example, System 2bba (Figure 4) has a penta-decapeptide ligand (Table 1) and a relatively extensive hydration structure of 27 water molecules in the extracted interface. In the case of 2bba, the largest decrease from 4.34 to 2.13 kcal/mol of the residual (COSMO models in Table 2) was obtained with Shell 1 water molecules. A detailed overview of the hydration structure shows that water molecules of Shell 1 (asterisks in Figure 4) mostly positioned at the bottom of the binding pocket and play a bridging role between the target and ligand partners. In this case, application of Shell 2 waters in addition to Shell 1 ones was not beneficial as they increased the residual. However, the final residual with Shell 3 is still below the dry model.

Beyond bridging and space filling roles presented in Figure 3 and Figure 4, interfacial hydration also exerts a shielding effect [66] on target-ligand intermolecular interactions, as well. Despite the importance of the hydration structure, crystallography often does not supply crucial water positions or erroneously assigns waters in close contact (see also Introduction). This leads to limitations of the use of experimental complex structures in drug design.

The present study has overcome such limitations of experimental determination of hydration structures, and calculation of ΔH_b_ was possible using complete interfacial hydration structures resulted exclusively by MobyWat calculations (see Methods). Besides hydration structures, missing ligand positions of four Systems (3ptb_pad, 3ptb_pam, 3ptb_pme, 3ptb_pmo) were also produced by computational modeling. Thus, modeling provided atomic resolution data reliably completing experimental structures and yielding robust correlations of the present study. Notably, modeling steps (Figure 5) of building the hydration structure and the full complex require only moderate computational resources, and can be accomplished on a single workstation. With the application of a parallelized MD engine and a supercomputing facility, the calculation time can be reduced to a couple of hours. The Fragmenter step takes some seconds.

## 3. Methods

### 3.1. Preparation of Complexes

The primary input structures of all systems (Table 1) were obtained from the Protein Databank (PDB [67]). All crystallographic water molecules were removed. Missing atoms of solute side chains (both protein and ligand) were reconstructed with Swiss PDB Viewer [68]. In the case of missing terminal and non-terminal amino acids, acetyl and amide capping groups were added with the Schrödinger Maestro program package v. 9.6 [69] to the N-and C-terminus, respectively. In cases of homodimer structures, chain A was selected for calculations.

### 3.2. Parameters of Non-Amino Acid Ligands

For non-standard (non-amino-acid) ligands or residues molecular mechanics force field parameters were obtained from the GAFF force field [70]. Considering a non-standard residue, it was first capped on both terminals, with Ace- and -NHMe groups and pre-minimized with PC Model 9 [71] using MMFF94 force field [72]. Subsequently, semi-empirical quantum mechanics optimization was performed with MOPAC-2009 [73] using the PM6 parameterization with a 0.001 gradient norm [74]. In all cases, the force constant matrices were positive definite. Then, the completely minimized molecules were uploaded to RED server [75] to perform ab initio geometry optimization to obtain partial charges by RESP-A1B charge fitting (compatible with the AMBER99SB-ILDN force field). The calculations were performed with the Gaussian09 software [76], using HF/6-31G* split valence basis set [77]. The caps on the termini were excluded from charge derivation, charge restraints were applied on these atoms. Normal mode analysis was performed using GAMESS [78] to ensure that the final geometry is in energy minimum. Bond stretching, angle bending, and torsional parameters were assigned with the parmchk utility of AmberTools 1.5 [79] and used together with the partial charges to build GROMACS [80,81] residue topology entries for the non-standard residues.

### 3.3. Calculation of Interfacial Hydration Structure

MobyWat [46] predictions along with GROMACS MD simulations were used for calculation of water positions in the target-ligand complex. A uniform procedure was followed based on Method 3 of a previous study [32] briefly described in the following points. An overview of the modeling steps described in the forthcoming sections is provided in a flow chart of Figure 5.

### 3.4. Molecular Mechanics Energy-Minimization during MobyWat Predictions

For pre-MD minimization, the target or complex structure was placed in a cubic box using a distance criterion of 1 nm between the solute and the box. Void spaces of the box were filled up by explicit TIP3P water molecules [82] with the standard gmx solvate routine of GROMACS. Counter-ions (sodium or chloride) were added to neutralize the system. A uniform, procedure was applied in all cases prior to the MD steps, including a steepest descent (sd) followed by a conjugate gradient (cg) step. Exit tolerance levels were set to 10^3^ and 10 kJ·mol^−1^·nm^−1^ while maximum step sizes were set to 0.5 and 0.05 nm, respectively. Position restraints were applied on solute heavy atoms at a force constant of 10^3^ kJmol^−1^nm^−2^. All calculations were performed with programs of the GROMACS software package [81], using the AMBER99SB-ILDN force field [83]. The above energy-minimization was performed twice, once for the target and once for the re-assembled target-ligand complex (see below).

### 3.5. Molecular Dynamics of the Protein Target

After energy-minimization, 5-ns-long NPT MD simulations were carried out with a time step of 2 fs. For temperature-coupling the velocity rescale [84] and the Parrinello–Rahman algorithm were used. Solute and solvent were coupled separately with a reference temperature of 300 K and a coupling time constant of 0.1 ps. Pressure was coupled by the Parrinello–Rahman algorithm [85,86,87] and a coupling time constant of 0.5 ps, compressibility of 4.5 × 10^−5^ bar^−1^ and reference pressure of 1 bar. Particle Mesh-Ewald summation was used for long range electrostatics. Van der Waals and Coulomb interactions had a cut-off at 11 Å. Coordinates were saved at regular time-intervals of 1 ps yielding 1.001 × 10^3^ frames. Position restraints were applied on solute heavy atoms at a force constant of 10^3^ kJ·mol^−1^·nm^−2^. Periodic boundary conditions were treated before analysis to make the solute whole and recover hydrated solute structures centered in the box. Each frame was fit to the original protein crystal structure using Cα atoms. The final trajectory including all atomic coordinates of all frames was converted to portable binary files. The target structure, and the surrounding (surface) water molecules were extracted as the last frame of the 5-ns-long MD simulation. At this point, there is a difference between the present study and Method 3 applied previously [32]. In Method 3, surface water molecules had been provided by MobyWat using 1-ns-long MD simulation. In the present study, the final frame of a 5-ns-long MD simulation was applied.

### 3.6. Re-Assembly of the Target-Ligand Complex

The target-ligand complex was re-assembled. For this, the target part of the holo and the hydrated apo systems were fitted on the top of each-other and the ligand was used together with the hydrated target (soaking), and interfacial water molecules were extracted. A water molecule was considered interfacial if intermolecular distance was smaller than/equal to a pre-defined maximal threshold (d_max_) of 5 Å for both the ligand and target partners. Water molecules conflicting with the ligand structure were excluded using the editing mode of MobyWat at a minimum distance limit (d_min_) of 1.75 Å prior the second MD simulation.

### 3.7. Molecular Dynamics of the Target-Ligand Complex

The MD simulation protocol described above for protein targets was performed for the re-assembled target-ligand complex structure, as well. In this case, all frames of the final trajectory of the target-ligand complex (in a water box) were used in the next step for production of interfacial water positions.

### 3.8. Production of Interfacial Water Positions

After the MD simulation of the target-ligand complex, MobyWat prediction of interfacial water positions was performed with dmax, clustering and prediction tolerances of 5.0, 3.0, and 3.0 Å, respectively. The MER clustering algorithm of MobyWat was applied. At this point, the present procedure differs from Method 3 [32]. As a result, a list of predicted water oxygen positions was produced by MobyWat in PDB format.

### 3.9. Molecular Mechanics Energy-Minimization after MobyWat

The MobyWat-supplied oxygen atoms of predicted water positions were equipped with hydrogen atoms and energy minimization was performed for the hydrated complexes. A four-step protocol was applied for energy minimization of complexes with predicted water positions following an sd-cg-sd-cg pattern with parameters of sd and cg methods described above. During the first two steps, all solute heavy atoms and the oxygen of the predicted interfacial water molecules were position restrained and bulk waters and ions were released. In the last two steps, position restraints were not applied on predicted waters, only solute heavy atoms were position restrained. Other details were the same as described in Section 3.4 above.

### 3.10. Extraction of Target-Ligand Interfaces by Fragmenter

Fragmenter automatically extracts target-ligand interfaces of large complexes and is freely available as a web service at www.fragmenter.xyz. Algorithm details and connections between input, algorithm, implementation, and output scripts are presented in Appendix A. In brief, the extraction is based on the selection determined by the target-ligand (d_TL_) and the water-solute (d_w_) distances as well as the inter-residual distance (n). In the main loop (Appendix A), a target amino acid residue is extracted if it has at least one heavy atom with d_cls_ ≤ d_TL_, where d_cls_ is the spatial distance measured between the closest heavy atoms of the actual target and ligand molecules. The maximal distance allowed between the closest heavy atoms of the target and the ligand (d_TL_) can provided by the user and a default value is set to 3.5 Å. The same distance between solute partners and water molecules (d_W_) is also defined and applied for extraction of interfacial waters. Connecting amino acids and terminating groups are also inserted. The length of fragment peptides is influenced by the maximal inter-residual topological distance (*n*) of the target. Parameter n specifies how many adjacent amino acids are added to the fragment chain of amino acids extracted above by the d_TL_ criterion. If *n* > 0, then the fragment was grown by adding n connecting amino acid residues. If *n* = 0 only amino acids with d_cls_ ≤ d_TL_ are added to the fragment chain. If *n* = 1, the sequential first neighbors are also attached to the terminus (termini) of the fragment chain, even if the attached amino acids have a d_cls_ > d_TL_, etc. (Appendix A).

#### 3.10.1. Input

The actual content of the query form of the ‘submit’ tab of the web interface (Appendix A) is saved as a single input file (project_ID.inp) generated according to a template inputfile.inp (Appendix A). This template contains the system variables, php path, the path of the createqinput.sh, and the template for the input parameters from the website. The ‘submit’ tab allows setting distance (d_TL_, d_W_, and n) and other parameters of Appendix A. Fragmenter offers an option to freeze (restrain) atomic positions by labeling certain groups of heavy atoms such as backbone C_α_-atoms, heavy atoms, all heavy atoms in the Mopac input file. The definition of these restraints, additional Mopac parameters, and other administrative details are also collected in the project_ID.inp file. The latter parameters include the path and the file name of the complex structure, the process name (for the SLURM workload manager), the path of the php executable and Fragmenter scripts the mopac license file (for SLURM) and the path of the mopac and php executable. Using the above path and file information, setting of system variables is performed by script genqinput.sh. The user does not need to care about server configuration (e.g., server specific php executable path), it is stored on the server. Clicking on the ‘submit’ button the script calculate.php checks the integrity of the complex structure (Appendix A) by a PDB to PDB file conversion using OpenBabel [88]. In the case of conversion errors Fragmenter terminates and the errors are displayed on a separate page. Then it collects and transforms the input parameters from inputfile.inp and from the site from the user for the script createqinput.sh and calls createqinput.sh. Script genqinput.sh requires only one input file (project_ID.inp), which contains all necessary parameters for the run (Appendix A).

#### 3.10.2. Main Algorithm

Having all input data in project_ID.inp, script createqinput.sh calls script fragment.php, the main engine of Fragmenter and creates the output files using other php classes (point.php, atom.php, charge.php, ligand.php). Among the classes (i) atom.php represents the atom objects with coordinates, type; (ii) module charge.php calculates the charge the ligand and generated fragment chains; (iii) module ligand.php handles a ligand object, contains the atoms, bonds, it reads and writes the pdb files; (iv) Point.php is a small class reserved for the coordinates of the atoms. Utils.php collects technical parameters, for example operating system dependent information, config file handling, etc.

Script fragment.php (Appendix A) includes steps for input processing, fragmenting, and working with output files. Target and ligand objects are obtained from the input steps, target, ligand residues, and water molecules are detected based on their chain IDs and residue types (WAT, SOL, H2O), respectively. Accordingly, the input structure is split into ligand, target and water molecules, the residues are sorted by their residue IDs and only heavy atoms of the target are examined. In the main loop of fragment.php, the target amino acid residues are selected according to dTL and n by ligand.php and point.php. Single residue-gaps are excluded by connecting two neighboring fragments by selecting the connecting residue, as well.

Having all target fragments produced in the previous steps, each of them are terminated by a uniform procedure as represented by the cycle of fragment.php (Appendix A). In the case of a free N-terminus a protonated amino group is built automatically by adding hydrogen atoms in a correct geometry. Similarly, in the case of a free C-terminus, a carboxylate anion is left unchanged. After merging ligand and water molecules with the target fragments into a new PDB file, the total charge of the complex is calculated and stored in the remark section of the file. For all cut target chains, Ac- (at N-terminus) and -NHMe (at C-terminus) blocking groups are built on both or non-free ends using atoms of previous and/or next amino acids of the chain and adding three hydrogen atoms to the methyl group (Appendix A). Following the generation of all fragments, the interface water molecules are extracted according to the intermolecular distance cut-off (d_w_, Appendix A). After extraction of the water molecules, their total net charges are calculated by charge.php using individual charges of amino acids (Appendix A) at pH 7. Special care was taken for disulfide bridges between side-chains of cysteine amino acids. Following the main loop (Appendix A), Cys residues connected via disulfide bridges are also selected and added to the fragments. Total net charge (Appendix A) of the target fragments is calculated. In the case of disulfide bridges or protonated sulfhydryl group the charge of Cys is automatically set to zero, otherwise −1. The charge of His is calculated according to the protonation state of the imidazole ring (−1, 0, +1).

#### 3.10.3. Target-Ligand Intermolecular Interaction Energy

Fragmenter calculates target-ligand intermolecular interaction energy (E_inter_) for the extracted interface, which is expressed as the sum of Lennard-Jones (LJ) and Coulomb potentials (Equation (5)). For both the LJ and Coulomb potentials, Amber force field parameters are used [83,89]. A per-residue list of the E_inter_ is printed in the ‘results’ table. The list can be used for identification of target residues colliding with the ligand as large E_inter_ values. In such cases, further MM energy-minimization may be required to achieve a complex structure appropriate for QM investigations.
(5)$$E_{\mathrm{inter}}=E_{\mathrm{LJ}}+E_{\mathrm{Coulomb}}=\overset{N_{T}N_{L}}{\underset{i,j}{\sum }}\left[\frac{A_{\mathrm{ij}}}{r_{\mathrm{ij}}^{12}}-\;\frac{B_{\mathrm{ij}}}{r_{\mathrm{ij}}^{6}}+\;\frac{q_{i}q_{j}}{4{\pi \varepsilon }_{0\;}\varepsilon_{r}r_{\mathrm{ij}}}\right]$$
$$\;A_{\mathrm{ij}}={\;\varepsilon }_{\mathrm{ij}}R_{\mathrm{ij}}^{12}\;;\;B_{\mathrm{ij}}=\;2\varepsilon_{\mathrm{ij}}R_{\mathrm{ij}}^{6}\;;\;R_{\mathrm{ij}}={\;R}_{i}+{\;R}_{j}\;;\;\varepsilon_{\mathrm{ij}}=\;\sqrt{\varepsilon_{i\;}\varepsilon_{j}}\;$$
where, ε_ij_ is the potential well depth at equilibrium between the ith (ligand) and jth (target) atoms; ε_0_ is the permittivity of vacuum; ε_r_ = 1, relative permittivity; R_ij_ is the inter-nuclear distance at equilibrium between ith (ligand) and jth (target) atoms; q is the partial charge of an atom; r_ij_ is the actual distance between the ith (ligand) and jth (target) atoms; N_T_ is the number of target atoms; N_L_ is the number of ligand atoms.

#### 3.10.4. Output

Fragmenter stores the output files in an output directory and provides a download link to all of them in the ‘results’ site (Appendix A). The ligand, the selected water molecules and the target fragments are downloadable as a complex in PDB format. The final charge of the complex is stored in the remark section of the PDB file. Fragmenter also provides additional separate PDB files and also converts them into downloadable Mopac input files. These files include the structures of the complex with/without water molecules, separate ligand, or target fragments. The ‘output’ tab also features the fragmented complex in a small window and it can be manipulated by the user. The visualization and rotation is performed by JSmol [90] implemented in the web page.

### 3.11. Calculation of Heats of Formation

Mopac 2012 [91] was used for structural relaxation and calculation of heats of formation of the extracted complex structures, separate ligand, water molecules, and target fragments. Hamiltonian of the Parametric Method number 7 (PM7 [92]) was applied. The exit criterion of the energy-minimization was defined as a gradient norm of 1.0. The value was set according to the instructions of the Mopac Support Team, and it is a magnitude smaller than the value of 10 suggested by the Manual [92]. There were only four vacuum calculations where the final gradient norm was slightly higher than 1, and the largest one of the four was 2.5. To reduce computational cost, the localized molecular orbital approach of Mozyme [93] was applied. Total net charges of the molecules were calculated from individual net charges of the amino acids (Appendix A). The charges were indicated in the command line, checked manually and automatically with keyword GEO-OK. To prevent unwanted termination of calculations, keyword PREC was applied. Eigenvector following [94] was used as a default geometry optimization. Molecular mechanics correction to peptide bonds was applied by keyword MMOK. Except the cases of in vacuo calculations, the COSMO (COnductor-like ScreeningMOdel) model [63] was used. For this, a value of 78.3 was set at the EPS key word which is the dielectric constant of water at 293.15 K and 101325 Pa. Δ_f_H of water was calculated with the above keywords in vacuum and using the COSMO model, respectively. In the cases of four systems, integrity of disulphide bridges was conserved by restraining the coordinates of S atoms during COSMO calculations.

### 3.12. Statistics

Simple linear regressions were performed between calculated Δ_r_H and experimental ΔH_b_(exp) values in all cases of Table 2 and Appendix A. ΔH_b_(exp) values were obtained from various publications as listed in Table 1. Statistical parameters of the regressions including regression coefficients (α and β in Equation (3)), coefficients of determination (*R*^2^), *t*-values, *F*-values, residuals and root mean square error (RMSE) values are listed in Table 2. Leave-one-out cross-validated R^2^ values were also calculated to check the stability of the correlations. Significance values of regression coefficients mentioned in the main text were calculated by two-sided *t*-test. For correlation plots, ΔH_b_(calc) values were calculated using Δ_r_H values and the regression coefficients (Equation (3)).

## 4. Conclusions

Structure-based calculation of binding thermodynamics is challenging at the QM level. To overcome the limitations of system size and hydration, a new protocol was introduced combining a MobyWat-based prediction of hydration structure with Fragmenter, a tool designed for extraction of the target-ligand interface with peptide fragments representing the target molecule. The protocol allowed fast QM calculations on a series of target-ligand interfaces with systematically adjusted hydration models. High correlations were achieved with a hybrid model involving a shell of explicit water molecules of calculated positions and the implicit solvation method COSMO. At semi-empirical QM level, and PM7 parameterization, a single, statistically significant scale factor was obtained for conversion of calculated reaction heats into experimental binding enthalpy values. The results of the present study will be particularly helpful in enthalpic optimization of drugs and in the molecular design of stable complexes and new ligands, in general. Further development and tests of the protocol have been also initiated for applications at the highest level of QM theory.

## Figures and Tables

**Figure 1 ijms-20-04384-f001:**
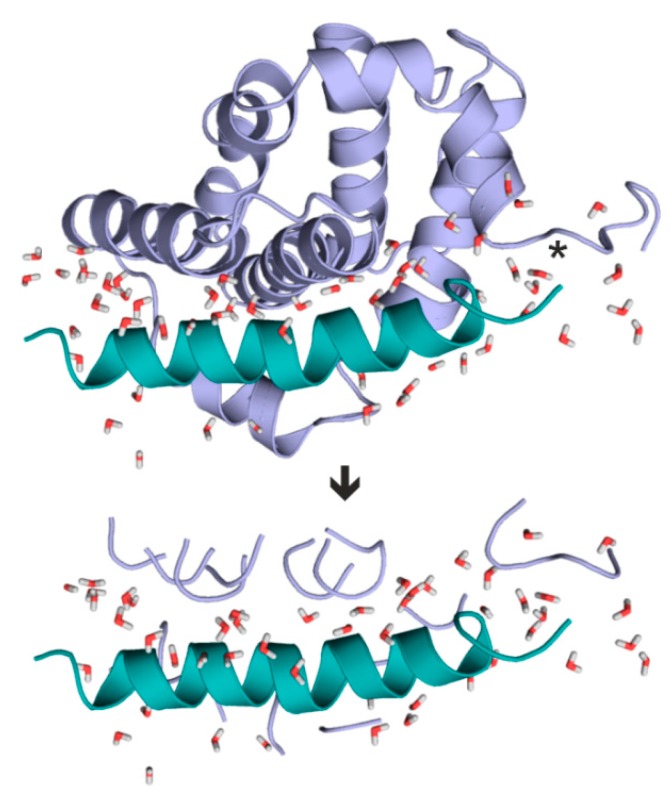
Fragmenter extracts a hydrated interface (bottom) from the target-ligand complex (top). Target (fragments) and ligand are shown in light blue, and green, respectively. System 2roc contains the largest ligand investigated in the present study. In this example, Fragmenter extracted target residues with (d_TL_ = 5.0 Å) considerably reducing the system size used for QM calculation. Interfacial water molecules (d_W_ = 5.0 Å, sticks) are also retained. Steps of extraction of target fragments are shown in atomic details for the C-terminal region (asterisk) in Appendix A as an example. Fragmenter is available free of charge as a web service at www.fragmenter.xyz.

**Figure 2 ijms-20-04384-f002:**
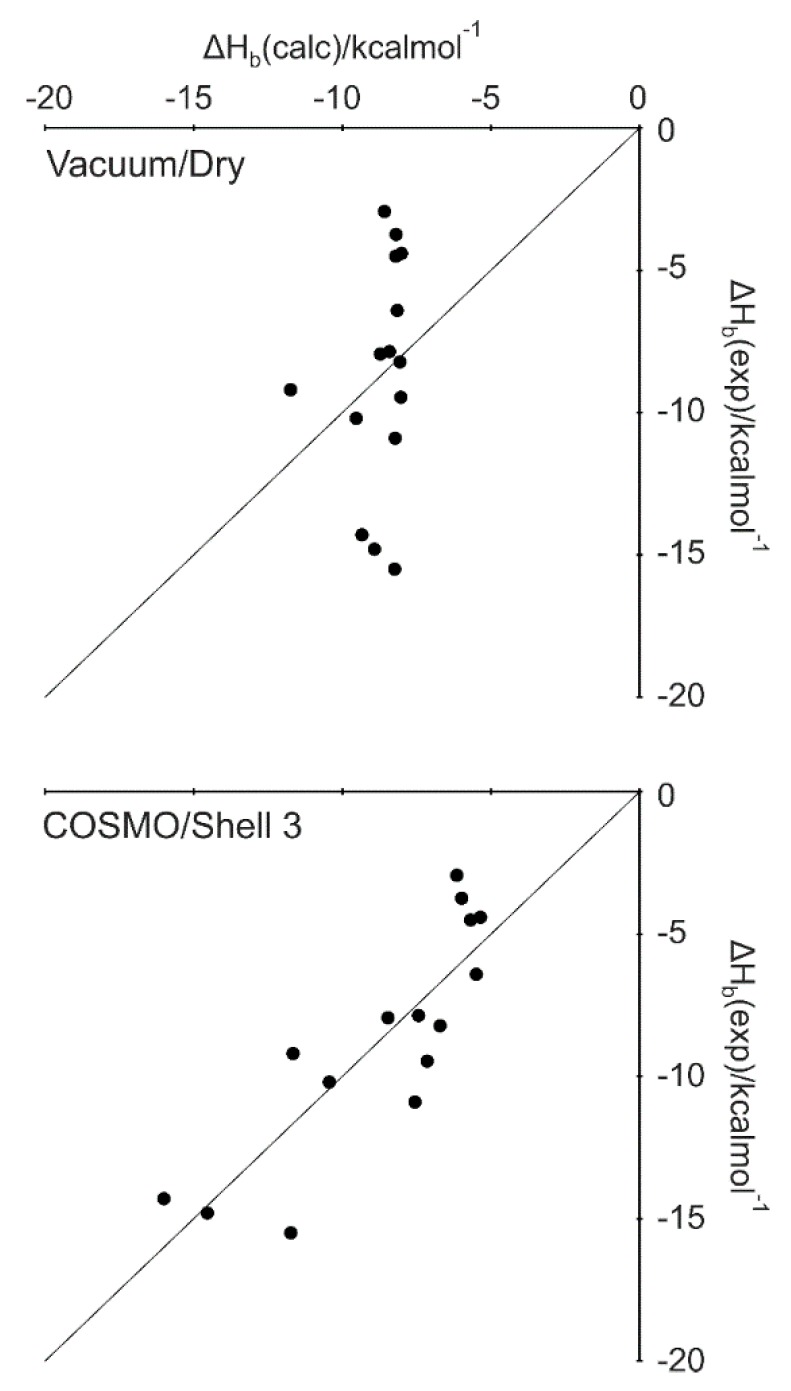
Correlation plots obtained without (Vacuum/Dry) and with (COSMO/Shell3) the hybrid water model.

**Figure 3 ijms-20-04384-f003:**
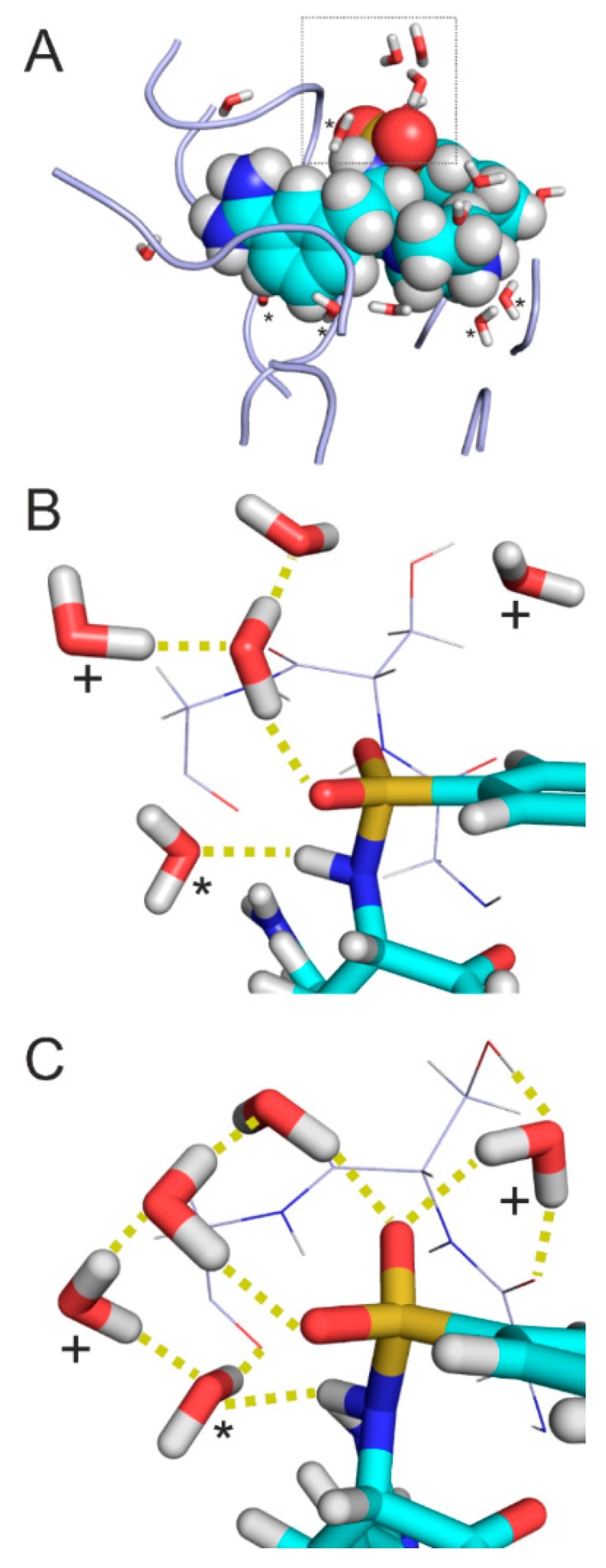
Extracted complex interface of System 1k1l. (**A**) Initial structure equipped with water molecules and energy-minimized at the molecular mechanics level. Target fragments and ligand are shown in ribbon and space filling representations, respectively. Water molecules in Shell 1 (d_W_ = 3.5 Å, sticks marked with asterisk) are positioned close to the solute partners and play a bridging role. The rest of Shell 3 (d_W_ = 5.0 Å) waters belong to Shell 2 (sticks without asterisks) and located at the edges of the interface, close to the bulk. Shell 3 = Shell 1 + Shell 2. (**B**) A rotated close-up of the box in Panel A showing the surrounding of the sulphonyl group of the ligand (sticks) and the neighboring residues G_216_SG_218_ of the target (lines) where the numbering follows that of the crystallographic structure (PDB ID 1k1l). Hydrogen bonds are marked with yellow dotted lines. (**C**) Structure in Panel B after relaxation at semi-empirical level using PM7 parameterization and Mozyme. Water molecules with a displacement above 1.5 Å after relaxation are marked with crosses.

**Figure 4 ijms-20-04384-f004:**
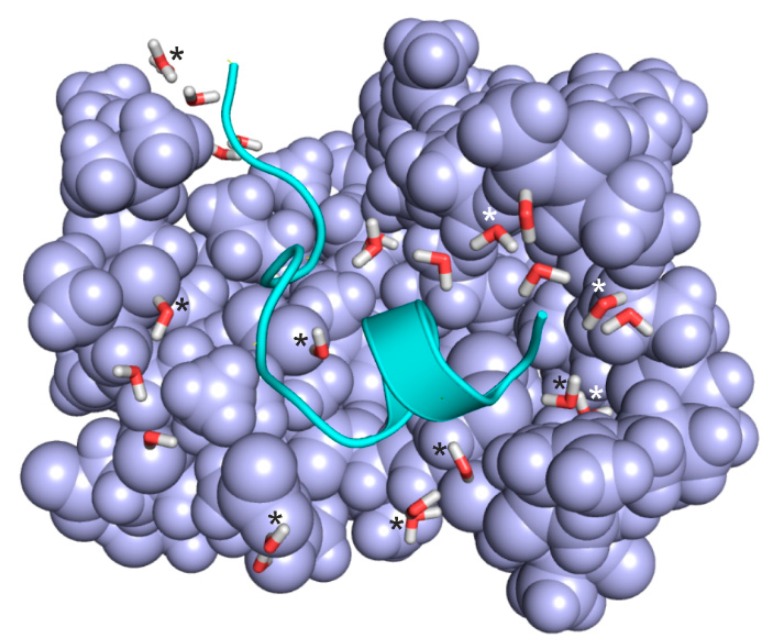
Extracted complex interface of System 2bba after relaxation at semi-empirical level using PM7 parameterization and Mozyme. Target fragments and ligand are shown in light blue space filling and green cartoon representations, respectively. Water molecules (sticks) in Shell 1 are marked with asterisks. Non-marked waters belong to Shell 2.

**Figure 5 ijms-20-04384-f005:**
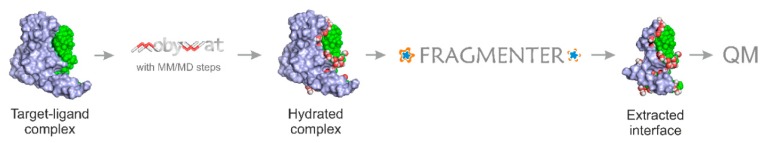
Modeling steps used for preparation of hydrated target-ligand interface for QM calculations shown on the example of System 1jgn. The procedure starts from the complex of the target (grey) and ligand (green) molecules. Program MobyWat [32,46] provides accurate hydration structure with MD-based calculations of positions of water molecules (red and white) in the interface. MobyWat is downloadable free of charge at www.mobywat.com. Finally, Fragmenter, a web service extracts the complex interface with considerably reduced target part for subsequent use in QM calculations. Fragmenter was introduced in the present study and available at www.fragmenter.xyz.

**Table 1 ijms-20-04384-t001:** Target-ligand systems.

System ^a^	Res ^b^ (Å)	Target	Ligand		Water Count			ΔHb(exp) ^d^
			Name	MW ^c^	Shell 1	Shell 2	Shell 3	kcal mol^−1^
3ptb_ben	1.7	beta-trypsin	benzamidine	121.2	1	6	7	−4.507 [2]
3ptb_pme	1.7	beta-trypsin	p-methylbenzamidine	135.2	1	5	6	−4.412 [2]
3ptb_pam	1.7	beta-trypsin	p-aminobenzamidine	136.2	3	4	7	−6.417 [2]
3ptb_pmo	1.7	beta-trypsin	p-methoxybenzamidine	151.2	1	6	7	−3.742 [2]
3ptb_pad	1.7	beta-trypsin	p-amidinobenzamidine	164.2	2	8	10	−2.935 [2]
1k1l	2.5	bovine trypsin	NAPe-piperazine	467.6	5	10	15	−7.863 [4]
1k1m	2.2	bovine trypsin	NAP ^e^−4-acetyl-piperazine	508.6	4	12	16	−8.222 [4]
1k1i	2.2	bovine trypsin	NAP ^e^-D-pipecolinic acid	508.6	2	13	15	−10.899 [4]
1k1j	2.2	bovine trypsin	NAP ^e^-isopipecolinic acid methyl ester	523.6	3	13	16	−9.465 [4]
1jyr	1.55	Grb2 SH2 domain	APS-PTR ^e^-VNVQN	1069.0	1	14	15	−7.94 [6]
1rlq	NA	C-src tyrosine kinase SH3 domain	RALPPLPRY	1084.3	2	25	27	−10.2 [7]
2ke1	NA	autoimmune regulator	ARTKQTARKS	1150.3	12	15	27	−9.2 [8]
2bba	1.65	EphB4 receptor	NYLFSPNGPIARAW	1606.8	12	15	27	−15.5 [9]
1jgn	NA	human poly(A)-binding protein	VVKSNLNPNAKEFVPGVKYGNI	2389.8	14	34	48	−14.8 [10]
2roc	NA	induced myeloid leukemia cell differentiation protein homolog	EEEWAREIGAQLRRIADDLNAQYERRM	3317.6	14	38	52	−14.3 [11]

^a^ System codes are derived from the PDB identifiers, and abbreviated ligands names (where applicable). ^b^ Resolution (available for crystallographic structures). ^c^ Molecular weight. ^d^ Experimental binding enthalpy values are given at their original level of precision except those with three decimal digits converted from kJmol^−1^, where 1 J = 4.184 cal. Sources of values are indicated as references in superscript. ^e^ NAP: N-alpha-(2-naphthylsulfonyl)-N-(3-amidino-L-phenylalaninyl); PTR: o-phosphotyrosine.

**Table 2 ijms-20-04384-t002:** Per-system residuals (ε) and statistical parameters of linear regressions obtained with different water models.

System	Vacuum				COSMO				
	Dry	Shell 1	Shell 2	Shell 3	Dry	Shell 1	Shell 2	Shell 3	Shell 3 ^b^
	|ε| ^a^								
3ptb_ben	3.70	3.18	2.90	1.93	3.73	2.33	2.45	1.18	0.85
3ptb_pme	3.60	3.09	3.05	2.03	0.53	1.12	2.48	0.95	1.22
3ptb_pam	1.74	1.29	0.88	0.03	0.01	0.02	0.44	0.92	3.03
3ptb_pmo	4.45	3.91	3.71	2.64	3.59	3.25	3.32	2.24	0.33
3ptb_pad	5.65	5.11	5.04	4.25	3.03	3.12	4.32	3.22	1.37
1k1l	0.56	0.39	0.10	0.10	2.59	1.05	0.56	0.43	1.74
1k1m	0.16	0.56	0.36	0.79	2.71	1.17	0.75	1.50	3.12
1k1i	2.67	3.19	2.95	3.51	2.80	3.62	2.88	3.34	4.60
1k1j	1.43	1.81	1.60	2.10	0.57	2.60	1.47	2.32	3.75
1jyr	0.78	0.28	0.53	0.09	1.73	0.36	0.40	0.53	0.36
1rlq	0.67	0.64	0.27	0.33	0.37	2.13	0.61	0.24	0.16
2ke1	2.54	4.67	4.66	5.28	4.38	4.21	5.73	2.46	2.89
2bba	7.25	6.38	7.21	6.23	4.34	2.13	6.58	3.77	3.31
1jgn	5.88	5.24	4.75	2.56	1.61	0.27	3.25	0.25	1.36
2roc	4.96	4.11	3.74	1.26	2.76	1.24	3.05	1.71	3.92
R^2^	0.06	0.18	0.19	0.44	0.51	0.65	0.33	0.73	0.93
R^2^(cv) ^c^	0.00	0.01	0.02	0.22	0.34	0.54	0.07	0.65	0.91
F	0.81	2.77	3.14	10.20	13.46	24.28	6.36	34.55	179.66
RMSE ^a^	4.02	3.76	3.72	3.10	2.90	2.45	3.40	2.17	2.65
t_α_	0.90	1.66	1.77	3.19	3.67	4.93	2.52	5.88	13.40
t_β_	−5.56	−5.04	−4.68	−3.90	−2.18	−3.99	−4.24	−2.81	-

^a^ Unit: kcalmol^−1^. ^b^ Linear regression with β = 0 (last column), and β ≠ 0 (other columns). ^c^ Leave-one-out cross-validated coefficient of determination.

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
