# Peer review of "A Fragmenting Protocol with Explicit Hydration for Calculation of Binding Enthalpies of Target-Ligand Complexes at a Quantum Mechanical Level"

_ijms, 2019, doi:10.3390/ijms20184384_

Round 1

Reviewer 1 Report

The submitted paper calculated the binding enthalpies of target-ligand complexes at a quantum mechanical level. The proposed topic is interesting, but there are some of the defects should be improved based on the suggestions below:

(1) Please change the subjective narratives become objective narratives in the manuscript. It is a research paper, in order to ensure the descriptions more scientific, the author should take care of that.

(2) Line 14, the abstract is a summary of the study, which should include the main objectives, the employed methodologies, the findings, and conclusion. Thus, I suggest the authors should re-write or deeply improve it.

(3) In the introduction section, it would be better the authors could add a sub-section for the descriptions of the objectives and the used methodologies based on the proposed literature reviews.

(4) Line 97, please separate the methodologies from the results and discussion. The present version or format of the study make the reviewer confused... How are the basic theories of the models? What are the innovations of the method you mentioned model? et al., please revise it. 

(5) Some of the problems in the building MD model, for example, for the energy-minimization of the proposed MD, how are many computing time you used? what is the calculation step in the model stabilizations or the calculations? It would be better for improving the readable of the paper for potential readers.

(6) I suggest the authors could add one calculation-flow for the models in the manuscript.

(7) I concerned that how is the author validated the model the author proposed??? The respective comparisons of the model results with the lab results or the published data.

(8) The conclusion section should be improved, it is so simple for the readers. 

In summary, the proposed research is useful, the main problem of the study is the confusing organization of the manuscript. It could consider for acceptance if the authors could improve it. 

Author Response

Response to Reviewer 1 Comments

The submitted paper calculated the binding enthalpies of target-ligand complexes at a quantum mechanical level. The proposed topic is interesting, but there are some of the defects should be improved based on the suggestions below:

Point 1: Please change the subjective narratives become objective narratives in the manuscript. It is a research paper, in order to ensure the descriptions more scientific, the author should take care of that.

Response 1: Subjective narratives were replaced by objective ones if detected throughout the text.

Point 2: Line 14, the abstract is a summary of the study, which should include the main objectives, the employed methodologies, the findings, and conclusion. Thus, I suggest the authors should re-write or deeply improve it.

Response 2: The abstract was re-written to provide a concise summary of the study. Methodologies (old and new ones) were explicitly mentioned, and the use of the results for enthalpic optimization was further emphasized.

Point 3: In the introduction section, it would be better the authors could add a sub-section for the descriptions of the objectives and the used methodologies based on the proposed literature reviews.

Response 3: A sub-section is added to the end of Introduction with descriptions of objectives and methodologies and theories including description of the end-point approach (Eqs. 1 and 2, formerly it was the first Section of Results and Discussion) with its literature references. The regression analysis (Eq. 3) part was move to Section 2.2.

Point 4: Line 97, please separate the methodologies from the results and discussion. The present version or format of the study make the reviewer confused... How are the basic theories of the models? What are the innovations of the method you mentioned model? et al., please revise it.

Response 4: Methodological details of Fragmenter (Section 2.1) are separated and moved to Methods Section 3.10. The new Section 2.1 is now focused on the discussion of the novel features of Fragmenter. Basic theory concerning Eqs. 1 and 2 is moved to the Introduction (see also Response 3).

Point 5: Some of the problems in the building MD model, for example, for the energy-minimization of the proposed MD, how are many computing time you used? what is the calculation step in the model stabilizations or the calculations? It would be better for improving the readable of the paper for potential readers.

Response 5: Both molecular mechanics energy minimization and molecular dynamics steps need relatively short computing times. The total time consumption of these steps can be largely reduced by massive parallelization of the runs. Models were considered stable once they reached the convergence criterion in Section 3.4 or the simulation time limit (Section 3.5) of a few nanoseconds as described in our previous studies (Refs. 32 and 46). To further improve the readability of the paper at this point a few sentences were added to the end of Section 2.5 on practical aspects of the model building procedures.

Point 6: I suggest the authors could add one calculation-flow for the models in the manuscript.

Response 6: According to the suggestion, a new flow-chart (Fig. 5) on the modelling steps was inserted in Methods.

Point 7: I concerned that how is the author validated the model the author proposed??? The respective comparisons of the model results with the lab results or the published data.

Response 7: Atomic resolution models of the complex systems are based on experimental structures with PDB identifiers listed in Table 1 (first column). Hydration structures of the complexes were built with a MobyWat-based methodology which was validated on numerous systems in previous works (Refs. 32 and 46). Experimental binding enthalpy data used for validation of the regression models of the present study were obtained from literature sources as listed in Table 1 (last column). Statistical parameters of Table 2 showed that the COSMO/Shell 3 model was the best performer as it was supported by the validation based on the experimental binding enthalpy data.

Point 8: The conclusion section should be improved, it is so simple for the readers.

Response 8: Section Conclusions was re-written and extended with additional details and future outlook.

In summary, the proposed research is useful, the main problem of the study is the confusing organization of the manuscript. It could consider for acceptance if the authors could improve it.

We thank the Reviewer for pointing to this weakness of the original manuscript. The above-mentioned considerable re-arrangements, re-writing of the text, and the suggested new organization hopefully increased the transparency and readability of the revised manuscript.

Reviewer 2 Report

In the article, a new protocol to improve the enthalpic design of new drugs has been introduced and validated via the comparison with measured enthalpies. The topic of the paper is important and of interest for a wide readership of IJMS. The key conclusions of the MS seem to be fully supported by  calculations, while improved correlations evidence the effeciency of the proposed approach. Overall, the paper is of good technical quality and posseses enough novelty to be published in IJMS after a few minor issues are addressed:
1. In the comparison of measured and computed enthalpies and the derivation of the scaling factor the impact of uncertanties in experimental enthalpies were neglected. What is a typical error in measured enthalpies/enthalpy change and how does this error impact the comparison of calculated and measured values and conclusions made based on the aforementioned comparison?
2. Large complexes considered here have very large number of isomers. What kind of configurational sampling was performed in the present study? How does its quality impact computed enthalpy changes?
3. The large hydrates studied here seem to be deeply inside the bulk region [Physical review letters, 96(12), 125701], at least thermodynamically, and thus, ΔHn,n-1 , where n is the hydration number for (A)(H2O)n complex, should be close to the bulk value for water. Is this the case? Please, comment.

Author Response

Response to Reviewer 2 Comments

In the article, a new protocol to improve the enthalpic design of new drugs has been introduced and validated via the comparison with measured enthalpies. The topic of the paper is important and of interest for a wide readership of IJMS. The key conclusions of the MS seem to be fully supported by  calculations, while improved correlations evidence the effeciency of the proposed approach. Overall, the paper is of good technical quality and posseses enough novelty to be published in IJMS after a few minor issues are addressed:

Point 1: In the comparison of measured and computed enthalpies and the derivation of the scaling factor the impact of uncertanties in experimental enthalpies were neglected. What is a typical error in measured enthalpies/enthalpy change and how does this error impact the comparison of calculated and measured values and conclusions made based on the aforementioned comparison?

Response 1: Uncertainties of measured binding enthalpies were reported as standard deviations for five of the fifteen original experimental papers. The values of standard deviations (SD) were 0.20, 0.24, 0.15, 0.60 and 0.10 kcal/mol for systems 1k1m, 1k1j, 1jyr, 1rlq, and 2bba, respectively. References of the experimental papers are listed in Table 1. In the case of the five 3ptb systems, a reproducibility of 5 % is mentioned in the Materials and Methods of Ref. 2. For the remaining five systems, explicit SD values were not found. Of these five systems, the measurements were done in duplicate (System 2ke1), standard deviations were not given after subtracting deprotonation enthalpy (Systems 1k1l, 1k1i), reason not mentioned (Systems 1jgn, 2roc). All-in-all, the error of reproducibility of ITC measurements is ca. 1-5 % (0.1-0.6 kcal/mol). As the above errors are relatively small, the measured binding enthalpy values are appropriate for a comparison with the calculated ones and expectedly do not have a remarkable influence on the conclusions.

Point 2: Large complexes considered here have very large number of isomers. What kind of configurational sampling was performed in the present study? How does its quality impact computed enthalpy changes?

Response 2: The present study is based on an end-point approach also applied by other papers listed in the Introduction. According to this approach, a single bound complex conformation is used for the calculation of binding thermodynamics, and therefore, configurational sampling was not applied for the solute (target and ligand) molecules. It is very important, that experimental conformations (for the PDB codes, please, refer to Table 1) were used for the solute molecules (expect a few systems where the ligand was modelled) which is undoubtedly a characteristic energy minimum conformation of the target-ligand complex. At the same time, for prediction of the hydration structure (which if often problematic for the measurements, see Introduction), extensive configurational samplings were performed using a series of MD calculations and the MobyWat program (Refs. 32 and 46) as described in Methods. In this way, reliable explicit water positions of the hydration shells were obtained, remarkably increasing the precision of prediction of binding enthalpy as it was described in Table 2 and the corresponding main text.

Point 3: The large hydrates studied here seem to be deeply inside the bulk region [Physical review letters, 96(12), 125701], at least thermodynamically, and thus, ΔHn,n-1 , where n is the hydration number for (A)(H2O)n complex, should be close to the bulk value for water. Is this the case? Please, comment.

Response 3: Yes, there are possible links between the above Phys Rev Lett paper and our work. For example, the approach and representation of formation of hydrate clusters in the Phys Rev Lett paper may be adopted for characterization of the role of individual water molecules in the hydration shells. The characterization and selection of influential water molecules would be particularly important in the design of new ligands. Maybe, ΔHn,n-1 could be used as an indicator of the influence/contribution of an individual water molecule to overall stability of the hydration shell and the target-ligand complex. In a forthcoming study, we plan to introduce such an indicator, and the Phys Rev Lett paper would be definitely an important literature prerequisite for that study.

We thank the Reviewer for careful evaluation and insightful comments on the manuscript.

Reviewer 3 Report

The authors present a fragment-based calculation protocol to obtain ligand binding enthalpies. The protocol is based on a sequence of MD simulations, prediction of interfacial water positions, MM energy minimization, and automated interface extraction. For the latter a web service is offered. Subsequently the heats of formation are calculated using the MOPAC program. The protocol is validated using a rather small set (15 values) of experimental binding enthalpies. The scaling factor between the heat of reaction (determined via the heats of formation) and the binding enthalpies is not well justified. Why is this factor required and what is the physical interpretation of its value? How general is it? It seems that the references cited by the authors (e.g. ref 54) do not use such a scaling factor. I therefore consider the manuscript of interest but too preliminary for publication. For practical (predictive) use a much larger validation set should be applied.

Author Response

Response to Reviewer 3 Comments

The authors present a fragment-based calculation protocol to obtain ligand binding enthalpies. The protocol is based on a sequence of MD simulations, prediction of interfacial water positions, MM energy minimization, and automated interface extraction. For the latter a web service is offered. Subsequently the heats of formation are calculated using the MOPAC program. The protocol is validated using a rather small set (15 values) of experimental binding enthalpies.

The scaling factor between the heat of reaction (determined via the heats of formation) and the binding enthalpies is not well justified. Why is this factor required and what is the physical interpretation of its value? How general is it? It seems that the references cited by the authors (e.g. ref 54) do not use such a scaling factor. I therefore consider the manuscript of interest but too preliminary for publication. For practical (predictive) use a much larger validation set should be applied.

Responses: Although a scaling factor (alpha) was not calculated explicitly in Ref. 54, it is common that a linear regression (Eq. 3) is performed between calculated and experimental thermodynamics values. The ranges of experimental and calculated thermodynamics values do not necessarily overlap (there is no such rule or law that they have to overlap) which may necessitate a linear scaling. At the same time, due to their correlation, the calculated binding enthalpies can predict the experimental ones. This is the case in our present study: a statistically significant linear relationship (Table 2, Eq. 4) was discovered between structure-based, calculated and experimental quantities. It is realistic to derive such a scaling factor between calculated and experimental quantities. It is very important that in Eq. 4 a simple scaling factor (alpha) was enough and the intercept (beta) was not necessary for establishment of the linear relationship. To mention an example from the literature, please refer e.g. to an excellent paper of Dobes et al. (J Comput Aided Mol Design (2011) 25:223–235) where several simple linear equations (with both alpha and beta) were published similar to Eq. 4 of the present study. The scaling factor of the present study does not depend on the energy unit and was derived for a wide range of ligands with MW up to 3300, and therefore, it has a general applicability for even large ligands. For the size of the validation set, please consider that similar number of systems were used in other relevant studies, as well. The number of systems with reliable thermodynamics data (especially binding enthalpy data) and structural information (PDB structure of the complex) at the same time is rather limited. Thus, for example Ref. 54 used 8 systems, Ref. 55 used 12 systems and the above paper of Dobes et al. used 15 systems. Thus, the number of systems (15) used in our present manuscript is similar to the afore-mentioned three studies published in reputable journals of the field and appropriate for publication.

The Reviewer is hereby acknowledged for careful evaluation of the study.

Round 2

Reviewer 1 Report

The quality of Figure 5 should be improved again. It is limited for the understanding of the potential readers. In addition, the authors have carefully revised the manuscript according to the suggestions from the reviewers.

Author Response

We thank the Reviewer for his/her positive comments on the revised manuscript. A new version of Fig. 5 is included in the manuscript with improved graphics and captions.

Reviewer 3 Report

The presentation of the material is much clearer in the revised version of the manuscript. I agree that reliable experimental data are scarce. As the authors provide access to their calculation protocol via a web server further evaluation by the community will be possible. 

Author Response

We thank the Reviewer for his/her positive comments on the revised manuscript.